# Therapeutic Targeting of Protein Lysine and Arginine Methyltransferases: Principles and Strategies for Inhibitor Design

**DOI:** 10.3390/ijms26189038

**Published:** 2025-09-17

**Authors:** Isaac Micallef, Byron Baron

**Affiliations:** 1Department of Tumor Genetics and Biology, Graduate School of Medical Sciences, Faculty of Life Sciences, Kumamoto University, Kumamoto 860-8556, Japan; isaac.micallef.17@um.edu.mt; 2Centre for Molecular Medicine and Biobanking, University of Malta, MSD 2080 Msida, Malta

**Keywords:** protein methylation, protein methyltransferases (PMTs), PMT inhibitors, methylproteomics, chemoresistance

## Abstract

Standard cancer chemotherapy is increasingly being supplemented with novel therapeutics to overcome known chemoresistance pathways. Resistance to treatment is common across various tumour types, driven by multiple mechanisms. One emerging contributor is protein methylation, a post-translational modification mediated by protein methyltransferases (PMTs), which regulate protein function by adding methyl groups, mainly on lysine and arginine residues. Dysregulation of protein lysine methyltransferases (PKMTs) and protein arginine methyltransferases (PRMTs) has been linked to cancer progression and drug resistance, making them attractive therapeutic targets. Consequently, several small-molecule PMT inhibitors have been developed, with some progressing to clinical trials. However, many candidates showing promise in preclinical studies fail to demonstrate efficacy or safety in later stages, limiting clinical success. This gap highlights the need to rethink current approaches to PMT inhibitor design. A deeper understanding of PMT mechanisms, catalytic domains, and their roles in chemoresistance is essential for creating more selective, potent, and clinically viable inhibitors. This review will summarise major chemoresistance pathways and PMTs implicated in cancer, then explore current and prospective PMT inhibitor classes. Building on mechanistic insights, we propose strategies to develop next-generation inhibitors with improved therapeutic potential against chemoresistant cancers.

## 1. Introduction

### 1.1. Challenge to Treatment

Chemoresistance is recognised as the greatest challenge in the successful treatment of cancer by chemotherapy, with the development and proliferation of highly aggressive phenotypes resulting in an associated poor prognosis.

Given the well-documented impact of methylation on protein expression levels through its action on histones, it is no surprise that protein methyltransferase (PMT) inhibitors have been considered as potential therapeutics to prevent or counter specific chemoresistance mechanisms, at least in a subset of patients, in the hopes of increasing the possibility of obtaining long-term benefits with the currently administered chemotherapeutic regimens. However, despite this knowledge related to histones, there is at present a lack of understanding as to the biological relevance of non-histone proteins targeted by PMTs as part of chemoresistance mechanisms.

The greatest stumbling block presently is that the functional role played by PMTs in either the development or the maintenance of chemoresistance is largely unresolved, preventing the rational design of targeted inhibitors. In the absence of clear mechanistic information, inhibiting the function of a PMT yields unpredictable clinical responses, where variability in efficacy and toxicity limits the safe inclusion of PMT inhibitors into combinatorial cancer therapy.

For this reason, effective design of PMT inhibitors requires understanding at least the core mechanism of methyl addition for the different PMT families and understanding their role in the major chemoresistance pathways, thus ensuring inhibitors are designed with high specificity and sensitivity for clinical application.

### 1.2. Chemoresistance Mechanisms

Across cancer types, therapeutic evasion is generally achieved by one or more of the following strategies employed by malignant cells. These mechanisms have been reviewed in detail elsewhere [1,2,3], but an overview will be provided in order to build the discussion in relation to PMTs.

From a genetic perspective, mutations that result in structural changes to the enzymes that the drugs interact with, such as alterations to catalytic sites that affect either drug binding or enzyme activation. For example, mutated topoisomerase II confers resistance to etoposide and doxorubicin [4]; mutations in the epidermal growth factor receptor (EGFR) kinase domain grant resistance against gefitinib [5]; mutations of the extracellular receptor smoothened [6] limit cyclopamine effectiveness; mutations in nucleoside transporters [7,8] block nucleoside drugs such as cytosine arabinoside (Ara-C, cytarabine) or 2-chlorodeoxyadenosine (2-CdA), whilst mutations in folate transporters [9] inhibit toxic folate analogues, such as methotrexate.

Another dominant mechanism is increased efflux of chemotherapeutic drugs, often mediated by either overexpression or increased activity of transporter proteins, including ATP-binding cassette (ABC) transporters. For example, increased expression of ABCB1 (P-glycoprotein, multidrug resistance 1) provides resistance to various anti-cancer drugs, including paclitaxel, olaparib [10], and doxorubicin [11,12]. The increased expression of ABCG2 provides resistance to doxorubicin [13], oxaliplatin [14], and 5-fluorouracil (5-FU) [15]. Similarly, ABCC2 (multidrug resistance protein 2; MRP2) confers resistance to cisplatin [16] and oxaliplatin [17].

Equally widespread among tumours is enhanced activity of detoxifying enzymes like cytochrome P450, which increases drug inactivation. For example, increased expression of glutathione transferases (GSTs) and production of glutathione (GSH) are effectively employed by tumours to detoxify many alkylating agents and platinum-based drugs [18].

Epigenetic and epiproteomic alterations are also frequently observed. Epigenetic changes are most commonly DNA methylation changes, including aberrant methylation of the ABCB1, ABCC1 and ABCG2 promoters to increase their expression [19,20,21], but can also involve post-transcriptional modifications through microRNAs (miRNAs) [22]. Epiproteomic changes encompass various histone modifications that alter gene expression patterns [23], including histone H3 hyperacetylation [24] or an enzyme such as histone deacetylase 2 (HDAC2) upregulating the expression of ABCB1 [25].

Other mechanisms include enhanced DNA repair capabilities [26,27,28], activation of other signalling pathways to bypass drug effects [29,30], and alterations in apoptotic pathways to evade cell death [30,31].

## 2. Protein Methylation

### 2.1. Types and Function

The cell has the ability to broadly modulate protein properties through the addition of post-translational modifications (PTMs), particularly methylation. These epiproteomic changes allow for fast cellular responses with minimal energy expenditure.

Protein methylation involves the transfer of a CH_3_ group from the methyl donor S-adenosylmethionine (SAM or AdoMet) to an amino acid residue on a protein. The residues altered are mainly lysine (K) and arginine (R) residues, and the methyl group is added onto nitrogen, making it N-methylation. A number of other amino acids can undergo N-methylation, namely asparagine (N), histidine (H), proline (P) and alanine (A). Two other types of methylation, S- and O-methylation, exist. The amino acids methionine (M) and cysteine (C) can undergo S-methylation, while aspartate (D) and glutamate (E) can undergo O-methylation [32,33,34].

One or more CH_3_ groups can be added to both K and R residues, with the epsilon (ε) amine of the former being able to undergo mono-(Kme1), di-(Kme2), or tri-(Kme3) methyl addition by protein lysine methyltransferases (PKMTs), while the guanidinium moiety of the latter can undergo mono-(MMA/Rme1), di-symmetrical (SDMA/sRme2) or di-asymmetrical (ADMA/aRme2) methyl addition by protein arginine methyltransferases (PRMTs) [35].

Multiple CH_3_ groups can be added to the protein before the protein dissociates from the PMT, which is referred to as a processive addition mechanism. On the other hand, if only a single CH_3_ group is added before the protein is dissociated from the PMT and subsequent methylations are added by the same or different PMTs, this is referred to as a distributive addition mechanism [36,37,38]. PKMTs and PRMTs implement both these mechanisms.

At a protein level, whilst methylation does not affect the overall charge of the residue, there is an associated increase in both hydrophobicity and bulkiness around the modified residue, which in turn can alter protein properties such as protein–protein interactions and recognitions, having a downstream impact on gene regulation [39,40].

In the context of cancer, the most well-studied substrates for methylation are histones, with exceptional focus on histones 3 and 4. Beyond histones, however, methylation has been identified on a wide range of non-histone proteins involved in transcription, DNA repair, RNA metabolism, and signal transduction on both arginine [37,41,42] and lysine [43,44,45] residues. Comprehensive proteomic analyses suggest that hundreds of cellular proteins undergo lysine or arginine methylation, involving crosstalk between PTMs, indicating a much more complex regulatory landscape than previously recognised [42,46,47]. Such modifications affect protein stability, sub-cellular localisation, and protein–protein interactions, thereby influencing pathways critical to oncogenesis and drug response. Importantly, methylation of non-histone substrates such as tumour protein 53 (p53), heat shock proteins 70 or 90 (HSP70 or HSP90), nuclear factor kappa B (NF-κB) and Retinoblastoma 1 (RB1) has been implicated in modulating DNA damage repair, apoptosis, and survival signalling in tumours [41,45]. Nevertheless, despite these modifications contributing to the emergence of chemoresistant phenotypes, the methylation status and functional roles of these key cancer proteins following chemoresistance remain poorly understood.

### 2.2. The Methyl Cycle: S-Adenosylmethionine Production

The primary methyl group donor for methylating various substrates, including DNA, RNA, proteins, lipids, and other molecules, is S-adenosylmethionine (SAM). SAM is synthesised from the conjugation of adenosine from adenosine triphosphate (ATP) with methionine. The reaction is catalysed by SAM synthetase (also known as. methionine adenosyltransferase; MAT), which dephosphorylates a molecule of ATP in the presence of potassium and magnesium ions [48], together with cleavage of the C5′-O5′ bond of ATP and changing the conformation of the ribose ring from C4′-exo to C3′-endo such that the sulphur of methionine makes a nucleophilic attack on the C5′ to form SAM [49].

Methylation reactions are catalysed by methyltransferases and release S-adenosylhomocysteine (SAH or AdoHC). SAH is a competitive inhibitor of MTases and is hydrolysed by adenosylhomocysteinase (AHCY) to homocysteine (Hcy). This reaction is reversible, meaning that increased Hcy levels can lead to the accumulation of SAH, which can bring about a lower methylation capacity. This can be represented as the SAM/SAH ratio [50]. Homocysteine is re-methylated to regenerate methionine using 5-methyltetrahydrafolate (5-MTHF) or betaine via betaine homocysteine methyltransferase (BHMT) [51]. 5-MTHF, produced from dietary folate (dihydrofolate) through the folate cycle, donates a methyl group, catalysed by methionine synthase (MTR) using vitamin B12 (cobalamin, Cbl) as its cofactor [52]. These steps all form part of what is called the methyl cycle (Figure 1) [53]. The essential nutrients driving this cycle are thus methionine, folic acid (for CH3-THF), and choline (to generate betaine).

Because SAM serves as the universal methyl donor, its dysregulation perturbs a wide spectrum of methyltransferase reactions, thereby modulating key metabolic pathways [54]. SAM has been shown to have anti-cancer properties, limiting proliferative, inducing apoptosis, and preventing metastasis, although the underlying molecular mechanisms remain unresolved [55,56]. It has also been shown to sensitise colorectal cancer (CRC) cells to 5-FU treatment via p-glycoprotein inhibition and a reduction in NF-κB acetylation/NF-κB protein ratio, suppressing NF-κB Activation [57]. SAM has been tested as both a diagnostic and prognostic biomarker in different cancers in the form of the SAM/SAH ratio and has been studied as a potential chemotherapeutic, both on its own and as a chemosensitising agent in combination with other drugs, in a wide variety of cancer types [58].

### 2.3. Protein Methyltransferases

Methyltransferases across all families, irrespective of their substrates, typically have catalytic domains consisting of a well-conserved SAM-binding pocket, together with highly variable substrate acceptor regions [36,59]. These two compartments are linked via a hydrophobic channel, allowing the transfer of a CH_3_ group from SAM to the target residue through an S_N_2 transition [60].

In the most basic arrangement, the Rossmann-like fold consists of regions of alternating β-strands and α-helices. This produces a flat β-sheet at the centre, with a set of helices on either side of the sheet. In such an arrangement, the N-terminal β-strand is located in the middle of the sheet, with the strand topology following the order 3-2-1-4-5-7-6, and having the 7th strand anti-parallel to the other strands. Furthermore, functionally important, conserved residues in such enzymes are often found either at the C-termini of the β-strands or else in the adjoining loops [61].

Despite their diversity, most PMTs can be clustered into three superfamilies, namely the Seven-β-strand (7βS) domain family, the SpoU-TrmD (SPOUT) domain family and the Su(var)3–9, enhancer of zeste (E(z)), and trithorax (trx) (SET) domain family based on their sequence similarities and conserved regions within their catalytic domains [34,44,62].

The 7βS family features a core structure containing two domains, a seven-stranded Rossmann-fold domain forming the binding site for the methyl donor SAM and a β-barrel for dimerisation [63]. The SET domain family has small β-sheets forming a knot-like structure, surrounded by a pre- and post-SET domain together with an iSET domain forming a helical structure adjacent to the substrate binding cleft [61]. The SPOUT methyltransferase family contain a distinctive α/β knot-like structure [34,64,65,66].

For the scope of this review the focus will be on the 7βS and SET domain families since the two major groups of PMTs, i.e., protein lysine methyltransferases (PKMTs) and protein arginine methyltransferases (PRMTs) fall within these families.

#### 2.3.1. Protein Lysine Methyltransferases (PKMTs)

Over 50 PKMTs have been identified in humans [45,62,67,68]. The PKMTs can be grossly subdivided into two groups: those enzymes containing a SET-domain and those that do not (i.e., class 1 canonical Rossmann-fold-like methyltransferases) [34,60]. PKMTs are more finely subdivided into eight groups (KMT1-KMT8) based on the sequence similarity flanking the SET domain, activity, and substrates [44,60,69,70]. However, for many KMTs the substrates remain unidentified, making it difficult to assign clear functional roles or to accurately classify their enzyme activity [67].

Those PKMTs having a SET domain present the typical small β-sheets that produce a knot-like structure, with a pre-SET, post-SET and iSET domains. The helical structure formed by the iSET domain contributes to the formation of a narrow channel connecting the SAM binding surface and the substrate pocket through which the amine of the lysine residue to be methylated on the substrate is oriented and can access the SAM [61]. All these domains are essential for enzymatic activity. Additionally, the SET proteins also contain four conserved sequence motifs, SET motif I (GXG), SET motif II (YXG), SET motif III (RFINHXCXPN), and SET motif IV (ELXFDY), where each motif is involved in the methylation reaction [60,71].

The non-SET domain KMT category is currently composed of disruptor of telomeric silencing-1-like (DOT1L) (which makes up KMT4) and a few members of the Methyltranferase-like (METTL) family [33,59,60]. DOT1L has a 7βS as part of its N-terminal catalytic domain, which contains seven conserved sequence motifs (I, II, III, IV, VI, VIII and X) as a methyl donor binding site and an active site pocket covered with conserved hydrophobic residues [71,72,73,74]. The lysine methylating METTL enzymes in this group include METTL10, METTL12, METTL13, METTL20, METTL21A, METTL21B, METTL21C, METTL21D, METTL21E and METTL22 [33,72,75,76].

#### 2.3.2. Protein Arginine Methyltransferases (PRMTs)

A total of nine PRMTs have been identified in humans. As part of the 7βS family, the PRMT family members share a common overall structure consisting of a methyltransferase domain, a 7-β-barrel Rossmann fold domain (that interacts with SAM) and a dimerisation arm [77]. In detail, PRMTs harbour a catalytic domain composed of six conserved sequence motifs, where motif I is at the core of the SAM-binding pocket with three highly conserved glycine residues (VLD/VGxGxG), post-motif I hydrogen bonds to SAM either through the use of an glutamic or aspartic acid residue (V/I-X-G/A–X-D/E), motif II forms a β-sheet that stabilises motif I (E/K/VDII), the ‘double-E’ motif positions the arginine substrate residue via two glutamic acid residues (SExMGxxLxxExM), motif III forms a parallel β-sheet with motif II (LK/xxGxxxP) and finally the THW loop that is essential for both substrate binding and N-terminal stabilisation [41,78]. 

These 9 enzymes are divided into three sub-types, with all three sub-types giving rise to MMA, but then Type I and Type II PRMTs subsequently proceed further to produce ADMA and SDMA, respectively [68]. The Type I group consists of PRMT 1, 2, 3, 4, 6, and 8, the Type II group is made up of PRMT 5 and 9, while Type III only includes PRMT7 [37,38].

Moreover, PRMTs have preferences for certain motifs flanking the arginine to be methylated. PRMT1, PRMT3, PRMT6, PRMT8 prefer glycine-arginine-rich (GAR/RGG/RG) motifs [37,78], PRMT4 (also named CARM1) prefers proline-glycine-methionine-rich (PGM) motifs [78], PRMT5 recognises RGG, RXR, and RG motifs [78], PRMT7 can recognise RXRXR or RXR motifs [37,60,78] and PRMT9 uniquely recognises the CFKRKYL motif in SAP145 [79].

#### 2.3.3. Cancer Dysregulation

The increased expression of PKMTs, particularly Euchromatic histone-lysine N-methyltransferase 2 (EHMT2), Enhancer of zeste 2 (EZH2), SET domain containing 7 (SETD7), SETD8, SET domain bifurcated histone lysine methyltransferase 1 (SETDB1), SET And MYND Domain Containing 2 (SYMD2) and SYMD3 as well as PRMTs including PRMT1, PRMT3, PRMT5 and PRMT6 in a variety of tumours of different types and origins has been shown to promote chemoresistance through cell cycle progression [80], cancer stem cell (CSC) renewal [81] and the alteration of key signalling pathways such as the Wnt, Notch or Phosphoinositide 3-Kinase/Protein Kinase B/Mammalian Target of Rapamycin (PI3K/AKT/mTOR) pathways [82,83,84]. The clinical evidence thus supports the potential of targeting selected PMTs as promising therapeutic targets [85,86].

### 2.4. Classes of Methyltransferase Inhibitors

Although PMTs are structurally diverse, they share a conserved architecture composed of two functional pockets, the SAM binding pocket and the methyl-substrate binding pocket. PMT inhibitors can thus be sub-divided based on the region of the enzyme they target and their mode of action (either covalent and non-covalent binders). More specifically, inhibitors can be designed to compete with the methyl donor, the methylation substrate or an essential cofactor, or alternatively bind at allosteric sites found either within the active or at alternative inactive states [63].

#### 2.4.1. SAM Competitive Inhibitors

SAM competitors or mimetics compete with the methyl donor for the SAM binding pocket of the PMT thus preventing the enzyme from functioning and adding the methylation to the target protein [87]. A few examples of the more successful SAM inhibitors due to their selective nature include the EHMT2 inhibitor BRD9539 and its methyl ester BRD4770 [88], the SMYD2 inhibitor PFI-5 [89], the EZH2 inhibitors EPZ6438 (tazemetostat) and SKLB1049 [90], the DOT1L inhibitors EPZ004777 and its derivative EPZ5676 (Pinometostat) [91], and the PRMT5 inhibitor LLY-283 and its derivative PF-0685580 [92].

Despite a large number of SAM competitors having been developed, unfortunately, many such inhibitors present limited specificity given that all methyltransferases use SAM as their methyl donor. Designing SAM competitive inhibitors is a challenge due to the high hydrophilicity of SAM requiring inhibitors to have a high enough polarity to be able to interact with the SAM binding pocket, whilst at the same time presenting sufficient hydrophobicity to be able to traverse cellular membranes in order to reach their target. However, the greatest limitation to such inhibitors is the high intracellular levels of SAM with which they must compete [93].

#### 2.4.2. Substrate Competitive Inhibitors

This class of inhibitors compete with the protein substrate to be methylated for the substrate binding pocket of the PMT. The development of substrate competitive inhibitors is a preferred strategy due to ample structural diversity within the substrate binding pocket of PMTs, allowing for increased specificity (given the selective nature of substrate recruitment) as well as having a lower polarity requirement compared to SAM competitive inhibitors [93].

A few examples of the more potent and promising substrate inhibitors are presented in Table 1.

#### 2.4.3. Bisubstrate Inhibitors

Inhibitors within this class occupy both the SAM and substrate PMT pockets. There are very few confirmed inhibitors within this class, with a few documented examples being the SET7/9 inhibitor set of AzaAdoMet substituted with various alkylamino groups [116], the PRMT1 inhibitor decamidine [117], the PRMT5 inhibitor CMP5 [118], and the PRMT6 inhibitor 6′-methyleneamine sinefungin (GMS) [119].

#### 2.4.4. Allosteric Inhibitors

This class of inhibitors impede enzyme activity by binding to sites on the PMT other than its two active site pockets. The best known of such inhibitors is the PRMT3 inhibitor SGC707. Since PRMT3 forms homodimers to perform its function, SGC707 binds to the pocket at the base of the dimerisation arm of the first monomer and the α-Y segment of the activation helix of the second monomer, inhibiting the formation of the PRMT3 homodimer as a result of its introduction [120]. One of the few other allosteric inhibitors available is the SMYD3 inhibitor diperodon, which binds to an allosteric site the same allosteric binding site, located within the C-terminus domain, adjacent to the active site containing domain [121].

#### 2.4.5. Complex Disrupting Inhibitors

Many PMTs form multi sub-unit complexes where the integrity of the complex is critical in order for the PMTs to perform their function. This opens the possibility of developing inhibitors that target regions other that the catalytic domain.

EZH1/2 must form part of the polycomb repressive complex 2 (PRC2) complex together with at least three other PRC2 sub-units in order to be enzymatically active. One of these sub-units is Embryonic Ectoderm Development (EED). Numerous inhibitors have thus been developed that bind to EED and disrupt the formation of the PRC2 complex [122]. A few noteworthy inhibitors include SAH-EZH2 [123], EED226 [124], A-395 [125], BR-001 [126] and EEDi-5285 [127].

Mixed-lineage leukaemia protein 1 (MLL1 also known as. Histone-lysine N-methyltransferase 2A—KMT2A) or MLL fusion proteins form a complex with the transcriptional regulators Menin or WD repeat-containing protein 5 (WDR5). Although these proteins are not required for MLL1 catalytic activity, they are still critical components of context-specific MLL complexes and as such can be targeted to indirectly block the activity of MLL. Menin interacts with the N-terminal region of MLL proteins [128], whilst WDR5 interacts with the WIN motif of MLL proteins [129]. Of note are the Menin inhibitors KO-539 (ziftomenib) [36] and M-808 [130], as well as the WDR5 inhibitors OICR-9429 [131] and DDO-2093 [132].

PRMT5 forms a complex with methylosome protein 50 (MEP50 also known as. WDR77 or p44), which acts as a co-factor, in order to achieve its methyltransferase activity [133,134] and this dependence upon a co-factor is unique among the PRMTs. Further proteins are recruited as adaptors to specify substrate selection including Chloride conductance regulatory protein (pICln), RIO kinase 1 (RioK1) and Cooperator of PRMT5 (COPR5) [135]. So far only one inhibitor has been developed to block the interaction between PRMT5 and MEP50—by interacting with the hydrophobic pocket within the TIM barrel of PRMT5 [136] whilst another has been developed to block the interaction between PRMT5 and RioK1 or pICln [137].

#### 2.4.6. Covalent Inhibitors

The vast majority of PMT inhibitors developed so far, irrespective of which region of the enzyme they bind to, tend to be reversible (non-covalent) inhibitors. Covalent inhibitors are designed to consist of a peptide sequence or chemical structure that can covalently bind to a nucleophilic amino acid that is not well conserved within or in the immediate vicinity of the PMT active site via a reactive group, called a warhead, which is generally an alkylating, acylating, phosphonylating, or sulfonylating functional group [138]. Targeting such a residue improves selective inhibition of the intended PMT over other family members [139]. Most covalent inhibitors are currently designed to target a non-catalytic cysteine residue through an acrylamide or other α,β-unsaturated carbonyl compound [138,140,141].

Targeting PMTs with covalent inhibitors offers several advantages compared to using non-covalent equivalents, as these often exhibit superior potency and biochemical efficiency due to their irreversible, non-equilibrium-based mechanism of action [139,142]. Unlike reversible inhibitors, covalent binders do not compete with either the ligands or SAM cofactor, enabling sustained target engagement and reduced dosing amount and frequency, thus reducing off-target effects. Their long residence time on the target translates into extended duration of action, which is especially beneficial when the protein of interest has a slow turnover rate [142]. A diversity of warheads are available for PMT covalent inhibitors targeting cysteine, which can be tailored to exploit the spatial and chemical reactivity of the active site, allowing for the design of smaller, more selective molecules without compromising potency, thereby improving drug-likeness. This is particularly important given that many drug candidates fail due to absorption, distribution, metabolism, and excretion (ADME)-related issues [139,140]. Moreover, covalent inhibitors can enhance target occupancy even at low plasma concentrations, offering improved therapeutic efficacy and an advantage in overcoming drug resistance [142].

A few noteworthy examples include the EZH2 inhibitor SKLB-03176 [143], the EHMT2 inhibitor MS8511 [144], the SETD8 inhibitor MS453 [112], the PRMT5 inhibitor hemiaminal 9 [145] and the PRMT6 inhibitor MS117 [146].

#### 2.4.7. PROTAC Inhibitors

More recently, targeted protein degradation has been applied to PMT inhibition through PROteolysis TArgeting Chimeras (PROTACs). These are bifunctional molecules consisting of a bait moiety that binds to the target protein and a moiety that binds an E3 ubiquitin ligase such as Von Hippel−Lindau (VHL) or Cereblon (CRBN) [147,148]. SAM-competitive, substrate-competitive, and allosteric binders can all be utilised as the bait moiety in PROTAC design. Once the PROTAC brings the target protein, it can recruit an E3 ubiquitin ligase, which is then able to polyubiquitinate the target. The polyubiquitin group signals the proteolytic degradation of the target protein by the ubiquitin-proteasome system (UPS).

The inhibition of methylation by PROTACs has so far only been implemented against a small number of PKMTs and PRMTs namely the PKMTs EHMT2, EZH2 and NSD2 and the PRMTs PRMT3, PRMT4 and PRMT5. As for the PKMT PROTACs, MS8709 [149], AMB-03-378 [150], G9D-4 [151] target EHMT2, MS8847 [152], MS8815 [153], MS177 [154], U3i [155], YM181 and YM281 [156], MS1943 [157], PROTAC E7 [158] target EZH2 and MS9715 [159], MS159 [160], LLC0424 [161], together with a handful of unnamed PROTACs [162,163], target NSD2 for degradation. For the PRMTs, there are only degrader “11” [164] for PRMT3, C199 [165] for PRMT4 and MS4322 [166], MS115 [167] and YZ-836P [168] for PRMT5.

Apart from targeting the PMTs directly, interacting sub-units of the PMTs within protein complexes have also been targeted for degradation resulting in the destabilisation of the entire complex. The two proteins targeted in this manner so far are EED as part of the PRC2 complex and WDR5 as part of the MLL (KMT2A) complex. The PROTACs designed to target EED are UNC6852 [169], PROTAC 1 and PROTAC 2 [170] and UNC7700 [171], while the ones targeting WDR5 are 8g and 17b [172], MS33 and MS67 [173], MS40 and MS169 [174] and MS132 [175].

In some cases, PROTACs have been reported to demonstrate greater potency than their parent conventional inhibitor. However, despite the promise displayed so far, there are still challenges in transitioning these molecules from in vitro to clinical testing, one of which is the development of resistance by tumour cells to the commonly used E3 ubiquitin ligase ligands VHL and CRBN through mutations or down-regulation of the ubiquitin ligase mechanism [176]. That being said, designing PROTACs against PMTs remains challenging [177], and the field is still in its early stages, with further optimisation required for effective development [178]. Moreover, since the choice of E3 ligase ligand in PROTACs affects selectivity and degradation efficiency, alternative, highly selective ligase ligands need to be sought to further optimise PROTAC design [179].

#### 2.4.8. Inhibitor Limitations

As outlined above, selective inhibitors are currently only available against a small sub-set of the known PMTs. This is due to the fact that the enzymatic domain is highly conserved. Segregating methyltransferases based on the structural diversity of their SAM-binding site produces two groups, with the first containing SET domain PMTs, whilst a second contains PRMTs, small molecule-, DNA- and RNA-methyltransferases, which all share a cofactor-binding Rossman fold. This clustering underscores the high likelihood of off-target effects when attempting to design an inhibitor that targets the SAM-binding site of a selected PRMT [63].

Although the relatively high sequence conservation at the SAM-binding site makes inhibitor design challenging, given that the ATP-binding pocket of kinases and the cofactor binding pocket of PMTs have been shown to present a similar level of structural diversity [87], and selective kinase inhibitors have been successfully developed, isoform-selective inhibitors could be designed for selected PMTs by exploiting the few variable positions available [63]. More selective inhibitors that target specific PMTs as potential therapeutics, especially for application in cancer combinatorial therapy, can be designed by leveraging advances in structure-based design and high-throughput screening [36].

Furthermore, most of the inhibitors developed thus far are reversible and despite the apparent benefits of covalent inhibitors, it is still difficult to identify adequate warheads to covalently bind to weaker nucleophilic amino acids such as lysine and methionine and since PMT targets do not always contain adequate cysteine, lysine or methionine residues within their active sites, targeting other nucleophilic residues such as tyrosine, serine, threonine, and histidine, is harder since these are less explored [139], and not so easy to target with the currently available warhead chemistry [140]. For this reason, it is important to optimise the effectiveness of currently available warheads (particularly acrylamide derivatives) as well as explore the development of novel electrophilic warheads in order to facilitate covalent PMT inhibitor discovery and produce potent and selective molecules that can meet the required clinical standards [139]. The investigation of targeting less commonly studied nucleophilic residues (i.e., lysine, tyrosine, serine, threonine, and histidine) may yield alternative options for the development of a new spectrum of covalent inhibitors [138,141].

Another challenge is the widespread presence and on-going development of pan-PMT inhibitors, which offer limited clinical benefit. These inhibitors not only target PMTs that may be essential for basic cellular function, making it difficult to predict the overall impact on tissues and organs, but often exhibit highly variable potencies across the PMTs they affect. Achieving a dosing balance that adequately inhibits all intended targets is extremely difficult, frequently resulting in toxicity.

## 3. Failure to Clinic

Most of the inhibitors developed so far are only useful for in vitro experiments since their specificity and sensitivity are not always up to the desired clinical standard. Moreover, even those inhibitors that do make it to clinical trials run the risk of becoming ineffective due to the development of resistance. Designing more effective inhibitors requires a clear understanding of the biochemical and physiological reasons underlying failures in preclinical or clinical trials.

Various reviews have listed the compounds that have entered clinical trials, their use, and failure point [180,181,182,183,184,185]. Unfortunately, so far, PMT inhibitors have failed to demonstrate sufficient therapeutic efficacy at non-toxic doses in clinical trials. Having a wide range of protein targets involved in numerous biochemical pathways leads to off-target effects, which bring about significant side effects, outweighing any tangible benefits. In essence, the beneficial therapeutic range, with acceptable side effect and toxicity levels in patients, is very narrow and as a result, it is a significant challenge to achieve the desired therapeutic effect [185].

Computational chemistry, through the implementation of molecular docking, combined with structural biology using PMT crystal structures, provides an invaluable tool set to guide drug design. The data collected from failed trials will provide a stronger basis for in silico prediction using high-throughput screening assays, including virtual screenings [178,180] to confidently shortlist the more promising candidates to be tested further in the wet lab.

## 4. Considerations for Designing Novel PMT Inhibitors

PMT inhibitors need to have sufficient potency, selectivity, and cell permeability to be successfully implemented in the clinic. Developing selective PMT inhibitors and improving their clinical efficacy will require approaches beyond incremental modifications of existing inhibitory molecules, with greater emphasis on exploring alternative inhibitory mechanisms. However, this is particularly challenging because all PMTs use SAM as the methyl donor and function through an almost identical mechanism to incorporate the methyl group. The fact that they share highly similar SAM- and substrate-binding pockets limits opportunities for designing selective inhibitors. In turn, targeting the non-catalytic domains of PMTs is still relatively unexplored, making the assessment of their chemical tractability a compelling area for further research.

Specifically to improve peptide-based inhibitors, one possibility is to replace the peptide backbone with peptide mimics, called peptoids or peptidomimetics (Figure 2). Peptide inhibitors suffer from poor stability and short half-lives due to proteolytic degradation. These peptoids consist of chains of N-substituted glycine residues, such that the side chain is on the nitrogen of the peptide backbone, instead of the α-carbon as in peptides. As a result of this difference, peptoids are less susceptible to degradation by hydrolysis, resulting in improved stability and longer half-lives as well as often having improved cellular permeability and bioavailability compared to their peptide analogues [138,186]. The success of this approach has been shown in the development of a peptidomimetic inhibitor against SETD3 [187] and a peptoid inhibitor against PRMT1 [186]. Peptidomimetics targeting SETD3 were designed based on a 16-mer β-Actin peptide (residues 66–81) with the histidine at position 73 (H73) being replaced by a panel of methionine analogues, which acted as substrate competitive inhibitors offered varying degrees of inhibition, with selenomethionine being the strongest inhibitor. These molecules showed that a 3-carbon core side chain seemed to be the optimal length and that apart from methionine and selenomethionine, inhibition of SETD3 could also be achieved by β-Actin peptides containing homomethionine, S-methylcysteine, leucine, norleucine, norvaline, and aminobutyric acid [187]. A peptoid that targets PRMT1 was constructed covering the first 16 residues of histone H4 and the guanidine on arginine at position 3 (R3) was replaced with a chloracetamidine group as a warhead, which is well-known to modify cysteine and used to target PRMT1 via cysteine at position 101 (C101) found within the active site, that is not conserved in most other PRMT family members such that it could bring about selective and irreversible modification of PRMT1 [186].

Despite the extensive literature on PMT inhibitors, the precise molecular mechanisms by which many of them engage their targets, such as binding kinetics, induced conformational changes, or effects on multi-protein complexes remain poorly defined, limiting efforts to design improved inhibitors. In order to make PMT inhibitor targeting safe for clinical application, the broad use of SAM by a wide variety of enzymes and the similarity between the active sites of enzymes within the same family pose significant challenges to overcome in PMT inhibitor specificity. Considering that each PMT has hundreds of substrate proteins fulfilling very diverse cellular roles, it is expected that perturbing the PMT concentration will have implications on numerous biochemical pathways, some of which may be essential and non-redundant for non-cancerous cells.

Given the growing repertoire of PMT inhibitors, there is a pressing need for comprehensive biochemical and biophysical characterisation, at least for the more successful and promising of the existing PMT inhibitors [60]. Demonstrating direct interaction between the inhibitor and PMT is imperative and can be achieved through methods such as isothermal titration calorimetry (ITC) or surface plasmon resonance (SPR). Generating structures of the enzyme–inhibitor complexes by using NMR or X-ray crystallography is essential to elucidate the mode of binding and how specificity is being achieved. Functionally important is the demonstration of selectivity of the designed inhibitors for targeting their intended PMT by interrogation against a panel covering a broad range of methyltransferases through robust cellular assays [39,40].

Inhibitor validation should begin with an in vitro assessment of the level of inhibition of methylation activity. This can be achieved either by quantifying methylated protein products or the by-product SAH, for which both direct or indirect approaches exist [188,189]. Radiometric assays and top-down mass spectrometry, often in combination with electrophoretic methods such as 2D-PAGE, enable direct quantification of methylated proteins. Alternatively, digested peptides may be analysed using bottom-up or middle-down mass spectrometry, often following antibody enrichment [188].

SAH quantification can be performed directly using anti-SAH antibodies or mass spectrometry. Indirect SAH-quantification methods include radioisotope-labelled assays (using SAM with a tritiated methyl group), a variety of enzyme-coupled spectrophotometric (colorimetric/fluorecence/luminescence) assays (such as reactive thio-mediated chromogenic assays, adenine/xanthine-mediated colorimetric assays, ATP-mediated luminogenic assays, fluorescence polarisation assays and fluorescence lifetime assays), antibody-based assays (namely enzyme-linked immunosorbent assays (ELISA) or dissociation-enhanced lanthanide fluoroimmunoassays (DELFIA)), proximity assays (in the form of proximity homogeneous assays or time-resolved fluorescence resonance energy transfer (TR-FRET) assays), microfluidic capillary electrophoresis, and reporter assays (either using chemical labelling of methyltransferase substrates or cell-based reporter assays) [188,189].

Ultimately, the most promising inhibitors will need to be evaluated in disease-relevant or animal models to demonstrate their bioavailability, cell permeability and target engagement within cells as well as their impact on upstream and downstream signalling pathways, thereby bridging the gap from in vitro potency to therapeutic relevance [40,188].

Both the mechanistic information on substrate docking and the interaction crystal structure provide key information that not only helps to better understand the mode of action and biological effects that these inhibitors have on their target PMTs but also to identify the strengths and limitations of the different design strategies. Such information should thus become a benchmark for any future inhibitors developed.

A particularly important segment of missing information is related to the impact that the dysregulated PMTs have on the mechanisms of specific chemotherapeutic drug classes. Since each cancer type is treated with a limited number of chemotherapeutics and generally has a handful of commonly dysregulated PMTs, knowing such information would be key towards providing more targeted treatment regimens and avoiding administration of ineffective medication [185].

Another aspect to consider is what the inhibitor is designed to target. In most cases, the inhibitor docking is designed against the canonical sequence and 3D shape; however this does not represent the active form of the PMT. Proteins can be active within cells as functionally different molecules stemming from a combination of genetic polymorphisms, RNA splice variants, and multiple PTMs, and this chemical diversity at a protein level produced from a single gene is collectively described as proteoforms [190,191,192]. This is particularly relevant to PMT inhibition because proteoform expression generally varies both spatially (across cells and tissues) as well as temporally (with age or over time, as well as after specific events such as disease), affecting both their abundances and interactions, following potentially altered sub-cellular localisation [191,193]. That being said, it depends very much on the PMT being targeted for inhibition, since some proteins (e.g., actin or histones) present hundreds of proteoforms, while others (and this is probably the category in which most PMTs fall) only present one or two active proteoforms [193]. For these reasons, it is important to investigate and identify which proteoform of a specific PMT needs to be targeted in a particular disease context, i.e., if it differs from the main proteoform found in healthy cells or other tissues and organs. This would provide a more targeted approach and increase the chances of success in the clinical setting.

As further efforts are made to transition such PMT inhibitors towards clinical application, it is important to also remember that cancer cells will eventually find ways of developing resistance to at least some of these inhibitors, and any prior biochemical and biophysical knowledge will come in handy to overcome such resistance. The best document case so far is for EZH2, where in lymphomas, resistance against inhibitors such as GSK126, EPZ6438 and UNC1999 was acquired through single point mutations at positions I109K, Y111L/N/D, Y661D, C663Y and Y726F, which prevented inhibitor binding [194,195,196] but could also be due to NSD1 loss (which mediated H3K36 dimethylation) [197].

Despite the acquisition of resistance against a specific inhibitor, treatment with an inhibitor that targets the same PMT through a different binding mechanism might be able to bypass the acquired resistance in at least some of the tumours [196]. Alternatively, a co-factor within the active complex might need to be targeted [124,125] or a different inhibition strategy needs to be adopted [170]. Combinatorial therapy blocking epigenetic crosstalk with acetylation (e.g., H3K27ac) and phosphorylation (e.g., Ribosomal S6 kinase 4 (RSK4) or Cyclin-dependent kinase inhibitor (CDKN) 1A/2A/2B) [198,199] and their mediated downstream oncogenic activation (e.g., phosphorylation of Extracellular signal-regulated kinase (ERK) or Aurora kinase B (AURKB)) [199,200] should also be considered for personalised medicine applications.

## 5. Conclusions and Future Perspectives

In order to gain a more comprehensive understanding of what the roles of at least the major PMTs implicated in cancer chemoresistance are, a collection of patient tumour data from cancer cohorts that are characterised at a protein level, not just genetically, will be essential. So far the literature mostly presents data related to the effects of inhibitors on PMTs dysregulated at the onset of tumour formation. Protein data collected before and after patient treatment with such PMT inhibitors would also provide information on off-target effects as well as key downstream outcomes linked to the combined action of the PMT inhibitor and the chemotherapeutic drug, allowing for improved design or dose administration adjustments. There is, as of yet, very limited literature related to PMT inhibitor efficacy, particularly in relation to countering chemoresistance.

In order to collect reliable data in this regard, there is a need to develop novel experimental techniques for determining PMT inhibitor efficacy, PMT expression quantification, and activity testing. This goes hand in hand with improved or new strategies for higher efficiency of protein and methyl-peptide enrichment, such as antibody-based immunoprecipitation, methyl-lysine/arginine affinity resins, or chemical labelling approaches to selectively capture methylated peptides [201]. Technological advances in quantitative proteomics, incorporating the use of stable isotope labelling by amino acids in cell culture (SILAC), tandem mass tag (TMT)/isobaric tags for relative or absolute quantitation (iTRAQ) labelling, and data-independent acquisition (DIA) mass spectrometry techniques are required that enable more sensitive and reproducible detection of PMT substrates and activity changes. This would allow for higher confidence PTM identification and quantification [32,56,202,203]. Complementary methods such as targeted multiple reaction monitoring (MRM) or selected reaction monitoring (SRM) further allow validation of site-specific methylation events in complex biological samples [202]. Over and above all this, it is expected that with the current trend of implementing artificial intelligence (AI)-driven design, inhibitory molecules with improved selectivity and pharmacological properties will be shortlisted by using AI to integrate structural data, high-throughput screening results, and proteomic datasets.

A particularly interesting and impactful future consideration involves the investigation of combinatorial therapies of PMT inhibitors with other small molecule therapeutics already in clinical use, given that tumours often employ multiple chemoresistance mechanisms in parallel. One strategy would be to pair PMT inhibitors with immunomodulatory molecules such as immune checkpoint inhibitors, since protein methylation by EZH2 in tumour-infiltrating regulatory T-cells contributes to immune evasion [204], and a number of preclinical studies have shown blocking lysine methyltransferases can enhance tumour immunogenicity via anti–Programmed Cell Death Protein 1 (PD-1)/Programmed Cell Death Ligand 1 (PD-L1) responses [205]. An equally compelling course of action involves combinations with signalling pathway modulators, particularly PI3K, Mitogen-activated protein kinase (MAPK), or Wnt pathway inhibitors (including via kinase inhibitors). EZH2 inhibitors are being tested clinically for synergy with androgen receptor antagonists, while both EZH2 and EHMT2 inhibitors are being administered in clinical trials with EGFR-tyrosine kinase inhibitors (TKIs) to assess for enhanced responses [206], including in resistant tumours. Such studies highlight the potential for PMT inhibitors to be combined with PI3K/AKT/mTOR or MAPK inhibitors, exploiting methylation-dependent regulation of transcription factors like signal transducer and activator of transcription 3 (STAT3) or NF-κB to overcome adaptive resistance. Finally, combining PMT inhibitors with DNA repair–targeting agents such as poly (ADP-ribose) polymerase (PARP) inhibitors may also prove effective, given the role of lysine and arginine methylation in regulating DNA damage repair proteins such as breast cancer type 1 susceptibility protein (BRCA1) and p53 [207,208]. Such data indicates that even if PMT inhibitors are unlikely to show great clinical potential alone, they can still be implemented as sensitisers that can enhance the effectiveness of existing cancer therapies on non-responding tumours.

## Figures and Tables

**Figure 1 ijms-26-09038-f001:**
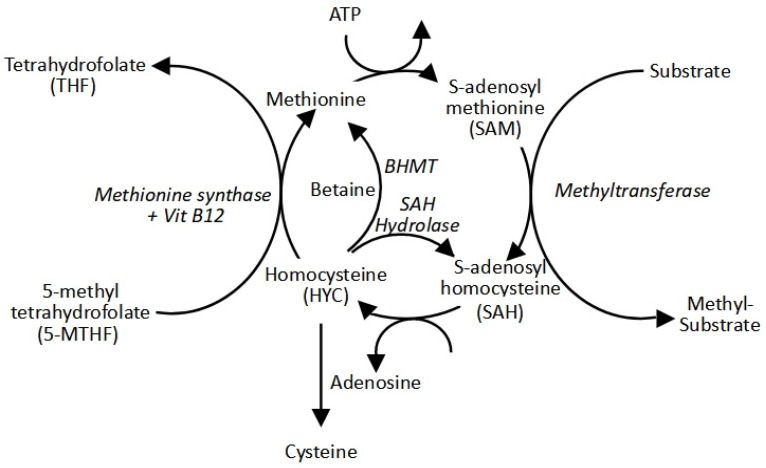
The methyl cycle.

**Figure 2 ijms-26-09038-f002:**
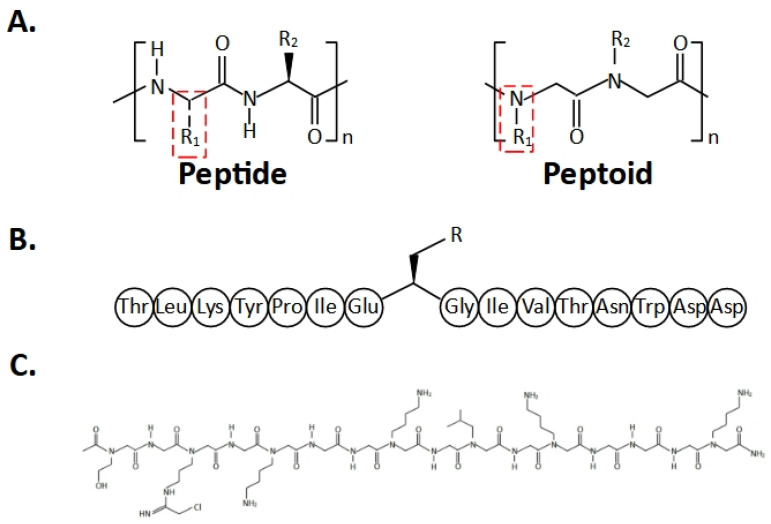
Peptidomimetic and peptoid inhibitors. (**A**) Structural difference between a peptide and a peptoid. (**B**) An example of a SETD3 16-mer peptidomimetic inhibitor based on the sequence of β-Actin residues 66–81 [187] (**C**) An example of a PRMT1 peptoid inhibitor [186].

**Table 1 ijms-26-09038-t001:** Noteworthy examples of substrate competitive inhibitors for some of the more popular methyltransferases.

Substrate	Inhibitor	Reference
EHMT2	UNC0642	[94]
	UNC0925	[95]
	A-366	[96]
	MS152	[97]
EZH2	UNC1999	[98]
	CPI-1205	[99]
	PF-06821497	[100]
	EBI-2511	[101]
SETD7	(R)-PFI-2	[102]
	Cyproheptadine	[103]
SMYD2	BAY-598	[104]
	EPZ030456	[105]
	A-893	[106]
	EPZ033294	[107]
SMYD3	BCI-121	[108]
	EPZ028862	[107]
	BAY-6035	[109]
SETD8	UNC0379	[110]
	SPS8I1	[111]
	MS2177	[112]
PRMT5	GSK3326595/EPZ015938 (Pemrametostat)	[113]
	GSK3235025/EPZ015666	[114]
	GSK3203591/EPZ015866	[115]

## Data Availability

Not applicable.

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
