# Peer review of "Therapeutic Targeting of Protein Lysine and Arginine Methyltransferases: Principles and Strategies for Inhibitor Design"

_ijms, 2025, doi:10.3390/ijms26189038_

Round 1

Reviewer 1 Report

Comments and Suggestions for Authors

The review article titled “Designing effective protein lysine and arginine methyltransferase inhibitors for chemotherapeutic targeting” presents an overview of protein methyltransferase inhibition in chemoresistant cancers. The authors explain the roles of lysine and arginine methyltransferases and provide an account of structural biology and inhibitor design across diverse mechanistic classes. This breadth and depth make it a valuable contribution. The discussion of challenges in developing selective inhibitors is strong, clearly linking laboratory obstacles with clinical translation failures. However, clinical relevance of specificity, toxicity, and resistance could be better connected to patient outcomes and supported by references to trial data.

The treatment of non-histone methylation substrates is brief, despite their importance in chemoresistance. Expanding this area would strengthen the review. Similarly, the call for improved proteomic and activity assays is valid, but more concrete methodological suggestions would make it more useful. Finally, clearer future directions, such as actionable strategies for targeted combinations (for example, PRMT5 combination therapies with Pluvicto, as discussed in the following abstract https://aacrjournals.org/cancerres/article/85/8_Supplement_1/590/755317) would improve the accessibility and impact of the current review. With stronger translational framing, this review could significantly enhance its impact on both researchers and clinicians. The authors need to revisit the title of the review and change it to the more appealing one (for example, Therapeutic Targeting of Protein Methyltransferases: Strategies for Designing Lysine and Arginine Inhibitors). Overall, this review could serve as a foundational resource on targeting PRMTs, offering useful recommendations.

Author Response

1) The treatment of non-histone methylation substrates is brief, despite their importance in chemoresistance. Expanding this area would strengthen the review.

We have now expanded the paragraph to read “In the context of cancer, the most well-studied substrates for methylation are histones, with exceptional focus on histones 3 and 4. Beyond histones, however, methylation has been identified on a wide range of non-histone proteins involved in transcription, DNA repair, RNA metabolism, and signal transduction on both arginine (Wei et al., 2014; Wesche et al., 2017; Al-Hamashi et al., 2020) and lysine (Lee et al., 2014; Zhang et al., 2015; Hamamoto et al., 2015) residues. Comprehensive proteomic analyses suggest that hundreds of cellular proteins undergo lysine or arginine methylation, involving crosstalk between PTMs, indicating a much more complex regulatory landscape than previously appreciated (Moore & Gozani, 2014; Biggar & Li, 2015; Wesche et al., 2017). Such modifications affect protein stability, sub-cellular localisation, and protein–protein interactions, thereby influencing pathways critical to oncogenesis and drug response. Importantly, methylation of non-histone substrates such as tumour protein 53 (p53), heat shock proteins 70 or 90 (HSP70 or HSP90), Nuclear factor kappa B (NF-κB) and Retinoblastoma 1 (RB1) has been implicated in modulating DNA damage repair, apoptosis, and survival signaling in tumours (Hamamoto et al., 2015; Wei et al., 2014). Nevertheless, despite contributing to the development of chemoresistant phenotypes in cancer cells, not much is known about the methylation status and functional role of these key cancer proteins following chemoresistance.”

2) Similarly, the call for improved proteomic and activity assays is valid, but more concrete methodological suggestions would make it more useful.

We have now expanded the paragraph to read “In order to collect reliable data in this regard, there is a need to develop novel experimental techniques for determining PMT inhibitor efficacy, PMT expression quantification, and activity testing. This goes hand in hand with improved or new strategies for higher efficiency of protein and methyl-peptide enrichment, such as antibody-based immunoprecipitation, methyl-lysine/arginine affinity resins, or chemical labeling approaches to selectively capture methylated peptides (Baron, 2021). Technological advances in quantitative proteomics, incorporating the use of SILAC, TMT/iTRAQ labeling, and data-independent acquisition (DIA) mass spectrometry techniques are required that enable more sensitive and reproducible detection of PMT substrates and activity changes. This would allow for higher confidence PTM identification and quantification (Deng et al., 2016; Levy, 2019; Micallef and Baron, 2023; Wang et al., 2017). Complementary methods such as targeted multiple reaction monitoring (MRM) or selected reaction monitoring (SRM) further allow validation of site-specific methylation events in complex biological samples (Micallef and Baron, 2023). Over and above all this, it is expected that with the current trend of implementing AI-driven design, there will be a push towards improving the currently promising molecules using AI to integrate data from various experimental inputs.”

3) Finally, clearer future directions, such as actionable strategies for targeted combinations (for example, PRMT5 combination therapies with Pluvicto, as discussed in the following abstract https://aacrjournals.org/cancerres/article/85/8_Supplement_1/590/755317) would improve the accessibility and impact of the current review. With stronger translational framing, this review could significantly enhance its impact on both researchers and clinicians.

We have now expanded the paragraph to read "A particularly interesting and impactful future consideration involves the investigation of combinatorial therapies of PMT inhibitors with other small molecule therapeutics already in clinical use, given that tumours often employ multiple chemoresistance mechanisms in parallel. One strategy would be to pair PMT inhibitors with immunomodulatory molecules such as immune checkpoint inhibitors, since protein methylation by  EZH2 in tumour-infiltrating regulatory T-cells contributes to immune evasion (Wang et al., 2018), and a number of pre-clinical studies have shown blocking lysine methyltransferases can enhance tumour immunogenicity via anti–PD-1/PD-L1 responses (Liao et al., 2023). An equally compelling course of action involves combinations with signalling pathway modulators, particularly PI3K, MAPK, or Wnt pathway inhibitors (including via kinase inhibitors). EZH2 inhibitors are being tested clinically for synergy with androgen receptor antagonists, while both EZH2 and EHMT2 inhibitors are being administered in clinical trials with EGFR-TKIs to assess for enhanced responses (Liao et al., 2023) including in resistant tumours. Such studies highlight the potential for PMT inhibitors to be combined with PI3K/AKT/mTOR or MAPK inhibitors, exploiting methylation-dependent regulation of transcription factors like STAT3 or NF-κB to overcome adaptive resistance. Finally, combining PMT inhibitors with DNA repair–targeting agents such as PARP inhibitors may also prove effective, given the role of lysine and arginine methylation in regulating DNA damage repair proteins such as BRCA1 and p53 (Zhang et al., 2022; O’Brien et al., 2023). Such data indicates that even if PMT inhibitors are unlikely to show great clinical potential alone, they can still be implemented as sensitisers that can enhance the effectiveness of existing cancer therapies on non-responding tumours.”

4) The authors need to revisit the title of the review and change it to the more appealing one (for example, Therapeutic Targeting of Protein Methyltransferases: Strategies for Designing Lysine and Arginine Inhibitors). Overall, this review could serve as a foundational resource on targeting PRMTs, offering useful recommendations.

We have edited the title to: Therapeutic Targeting of Protein Lysine and Arginine Methyltransferases: Principles and Strategies for Inhibitor Design

Reviewer 2 Report

Comments and Suggestions for Authors

The presented review paper is devoted to the discussion of potential usage of inhibitors of protein methyltransferases in anticancer therapy. The paper seems relevant to the scope of the journal and the amount of the data is sufficient for publication. However, there are several issues to be addressed:

  1. Table 1 with the examples of substrate competitive inhibitors for several methyltransferases should be broadened with structures of the inhibitors and the inhibitor classes.
  2. The Table summarizing the protein methyltransferases and their target genes should be added to the sections 2.3.2. Protein Arginine Methyltransferases(PRMTs) and 2.3.1. Protein Lysine Methyltransferases (PKMTs).
  3. The Table with the clinical trials described in the literature, their outcomes and disadvantages should be added to the section 3. Failure in Clinic.
  4. The Figure illustrating and summarizing the sequences and structures of peptoids or peptidomimetic should be added to the section 4. Considerations for designing novel PMT inhibitors.
  5. The extensive English editing should be performed. The text should be revised to be more concise, more logic, better organized.
  6. Typos, repeats, the sentences without the specific sense should be eliminated.
  7. Text formatting should be unified. The reference style should be edited according to the journal requirements.
  8. Abbreviation list is missing.
Comments on the Quality of English Language

The extensive English editing should be performed. The text should be revised to be more concise, more logic, better organized.

Author Response

  1. Table 1 with the examples of substrate competitive inhibitors for several methyltransferases should be broadened with structures of the inhibitors and the inhibitor classes.

We apologise but there is no space for the structures to be included at a decent size within the space available, and in any case most of our readers don’t care what the molecule looks like. The inhibitor class is “substrate competitive” for all as mentioned in the Table caption.

  1. The Table summarizing the protein methyltransferases and their target genes should be added to the sections 2.3.2. Protein Arginine Methyltransferases(PRMTs) and 2.3.1. Protein Lysine Methyltransferases (PKMTs).

There are 9 PRMTs and over 50 PKMTs with 10s if not 100s of targets each. It is beyond the scope of this review to list such a massive list of target proteins when we do not build further on their identity and do not mention them in the text.

  1. The Table with the clinical trials described in the literature, their outcomes and disadvantages should be added to the section 3. Failure in Clinic.

Same as above, there is a very long list of clinical trials (most of which were unsuccessful) and there already are entire reviews listing such clinical trials (and we reference a few of them). We do not build further on the specific outcomes of clinical trials and do not mention them in the text so it is beyond the scope of this paper to list clinical trials.

  1. The Figure illustrating and summarizing the sequences and structures of peptoids or peptidomimetic should be added to the section 4. Considerations for designing novel PMT inhibitors.

We have added a Figure (Fig 2) to show the sequences and structures of peptide vs peptoid and gave 2 examples mentioned in the text

  1. The extensive English editing should be performed. The text should be revised to be more concise, more logic, better organized.

We have gone through all the text again to ensure the English language is correct and sentences are clear.

  1. Typos, repeats, the sentences without the specific sense should be eliminated.

Any typos and repeats have been eliminated and sentences without the specific sense have been reworded

  1. Text formatting should be unified. The reference style should be edited according to the journal requirements.

The text formatting has been re-checked. The references will be edited according to the journal requirements once the reviewing is complete and no more references are added or removed.

  1. Abbreviation list is missing.

Full abbreviation list now added

Round 2

Reviewer 2 Report

Comments and Suggestions for Authors

No further comments